# Classifications of Sustainable Manufacturing Practices in ASEAN Region: A Systematic Review and Bibliometric Analysis of the Past Decade of Research

**Muhammad Imran Qureshi** [1,*], **Nohman Khan** [2], **Shazia Qayyum** [3], **Subha Malik** [4], **Sanil S Hishan** [5] **and Thurasamy Ramayah** [6,7,8,9]

1. Faculty of Technology Management and Technopreneurship, Universiti Teknikal Malaysia Melaka, Durian Tunggal 76100, Malaysia
2. UniKL Business School Universiti Kuala Lumpur, Kuala Lumpur 50300, Malaysia; nohman.khan@s.unikl.edu.my
3. Institute of Applied Psychology, University of the Punjab, Lahore 54590, Pakistan; shazia_agha@hotmail.com
4. Department of Gender & Development Studies, Lahore College for Women University, Lahore 54000, Pakistan; Subhamalik@yahoo.com
5. Azman Hashim International Business School (AHIBS), Universiti Teknologi Malaysia, Kuala Lumpur 54100, Malaysia; hishanssanil@gmail.com
6. School of Management, Universiti Sains Malaysia, Minden, 11800 Penang, Malaysia; ramayah@usm.my
7. Department of Management, Sunway University Business School (SUBS), Selangor 47500, Malaysia
8. Faculty of Economics and Business, Universiti Malaysia Sarawak, Kota Samarahan 94300, Sarawak, Malaysia
9. Faculty of Accounting and Management, Universiti Tunku Abdul Rahman (UTAR), Cheras, Kajang 43000, Selangor, Malaysia
* Correspondence: qureshi@utem.edu.my

**Abstract:** This paper aimed to map the existing sustainable manufacturing literature to explore and classify existing practices to highlight the potential prospect of and obstacles to achieving manufacturing sustainability in countries in the Association of Southeast Asian Nations (ASEAN). This paper systematically reviews the research on sustainable manufacturing in the ASEAN region from 2011 to 2020. We used the PRISMA framework for systematic literature, and 118 research articles specific to the ASEAN region were identified through a structured keyword search in Web of Science, SCOPUS, ProQuest, and other databases. After a careful screening process, only 115 records were found appropriate to be included for review in the current study. The results revealed three significant sets of manufacturing practices that are widely used for sustainable manufacturing. These are sustainable product development, sustainable manufacturing performance, and environmental assessment and monitoring. Furthermore, we conducted a bibliometric analysis of the literature to highlight ASEAN countries' collaborative efforts to achieve sustainability in manufacturing. The findings indicate that most of the earlier work on sustainable manufacturing focused on environmental assessment practices rather than providing holistic industrial engineering solutions. We recommend that the efforts focus on hybrid processes to establish sustainable manufacturing procedures in ASEAN member countries. Holistic solutions through industrial processing integration need to be developed to provide broader industrial solutions to protect the environment and society from the adverse effects of the manufacturing process with economic efficiency.

**Keywords:** sustainable manufacturing; triple bottom line; sustainability assessment; manufacturing industry; sustainable product development

## 1. Introduction

New environmental issues such as biodiversity, resource depletion, and climate change were important to developed countries' political agenda in the 1990s and early 2000s. In fact, these widespread environmental issues are more complex and differ in intensity from the existing ones. These issues are primarily attributed to industrialization, such as acid rain, water pollution, waste issues, and air pollution. Although initial environmental problems have been addressed impartially through clean technologies, new ecological issues require more fundamental transitions in the coming decades, i.e., significant changes in transport, energy, and agricultural systems due to the severe nature of upcoming environmental issues [1].

The discussion about sustainability generally starts with Brundtlan's [2] definition: "meeting the requirements of the current deprived of disturbing the capability of upcoming generations to encounter their desires." According to sustainability principles, sustainable manufacturing is the discovery of industrial products through a procedure that lessens harmful environmental effects, vibrant energy, and natural resources. Simply, this is safe for workers, societies, and clients and is economically feasible. Sustainable manufacturing includes many exercises to deliver improved environmental protection by administrations. Manufacturers are considering the ways to enhance market share at the same time by replying to essential global ecological issues. Hence, it is significant for the business sector to exceed budgetary requirements to achieve ecological objectives [3].

The sustainable manufacturing continuum calls for a complete view of the entire manufacturing supply chain with product lifecycle influence on financial, ecological, and social magnitudes [4]. Industrialized businesses identify the widespread advantages of sustainable manufacturing, accepting its assistance in improving cost [5], financial performance, and marketplace presentation, and delivering competitive leads [6]. It is important to develop rule and regulations within the organizations in order to receive competitive advantage through sustainable manufacturing.

The manufacturing sector is the most resource-consuming sector of the economy. Consequently, manufacturing activities utilize a considerable amount of energy and natural resources [7]. The International Energy Agency declared that 36% of carbon dioxide ($CO_2$) emissions worldwide are due to the manufacturing sector [8]. Similar to the case of countries in the Association of Southeast Asian Nations (ASEAN), economic development goals cause a threat to the environment. However, the improvement potential for sustainable development is significant. Technological changes have decreased the rate of $CO_2$ emissions. ASEAN countries encourage nuclear energy by ensuring safety and safeguards under international standards. ASEAN members also agree to increase collaboration on the joint research and development of low emission expertise for the cleaner use of hydro-carbon fuels and thus to recognize that fossil fuels will still play a vital part in sustainable manufacturing. Still, the biggest hurdle to achieving green growth is the production process in large manufacturing industries.

Typically, studies of manufacturing production are mostly attentive on refining competence and plummeting price. However, recently, with the growing responsiveness of severe sustainability-linked matters such as global climate warming and change, the switch and decrease in environmental and social influences have become a different objective in manufacturing industries [9]. The manufacturing sector has a crucial impact on development [10].

Sustainable development is primarily affected by manufacturing. In a growing atmosphere, sustainable manufacturing favors the existence of several organizations. The greatest common definition of sustainable growth is "fulfilling current requirements without compromising the capacity of future needs." The reasons behind the growing interest in sustainability are inadequate resources [11], overpopulation [11], poverty and industrialization [12], declining existing values, poisoned natural resources, global climate change, and an increase in the use of natural resources and concern for biodiversity and ecosystems [13]. Such subjects generate difficulties in accomplishing optimal development, which delays corporations' profitability. Sustainability comprises the philosophies of environmental friendliness, economic development, and social impartiality within sustainability [14]. Management, supply chain, manufacturing procedures, processes, and manufacturing fields are careful

to apply the sustainability model for enhanced output [15]. It contains the alteration from standard engineering methods to the contemporary progressions that consider their outcome on the economy, atmosphere, and culture [16]. Sustainable manufacturing talks about the problems connected to the strategy of engineering procedures and the implementation of the latest professional equipment [15,17]. Sustainable processes also protect advanced supply plans with determined economic earnings [18]. Goods and procedures as outcomes of sustainable manufacturing have no dangerous effect on human well-being and society [19]. Sustainable manufacturing also maximizes firms' new opportunities concerning new product development and further marketplace expansion [20].

Environmental protection was a central agenda in most of the summits of the Association of Southeast Asian Nations (ASEAN). ASEAN member countries are keenly focused on the adaptation of sustainability in manufacturing processes through collaborations, and their list of summits is raising the point to deliver. It is equally important to map achievements of the objective posted in ASEAN summits. Therefore, there are two main objectives of this paper. The first is to map existing sustainable manufacturing practices in ASEAN countries. The second is to assess the collaborative research work among ASEAN countries to achieve the common goal of green growth.

## 2. Review Methodology

This paper followed the PRISMA framework for reviewing existing literature [21]. As mentioned in the PRISMA guidelines, the scoping procedure was used to extract the most relevant articles on sustainability indicators and assessment. This practice facilitated the regulation of the critical lessons' obligatory features and classified the possible search keywords [22].

Due to the manufacturing practices' broader nature, a thorough literature review was performed using multiple databases to find relevant scientific journals and articles. Several keyword combination searches were conducted to collect the relevant published articles from the most renowned and credible research databases. These databases were Web of Science, Scopus, ProQuest, Science Direct, and EBSCO. The keyword "sustainable manufacturing" was used most in each database search to find the relevant literature. The data search was refined using predefined inclusion and exclusion criteria and quality standards. Each filter ensured the quality standard, and inclusion and exclusion measures are discussed in the next section.

The literature search's timeframe covered the past decade (2011–2020) to ensure current sustainability practices were highlighted. At first, a total of 1271 documents was shown; however, this included all types of documents, e.g., research articles, reviews, editorials, book chapters, and others. We limited our literature search to research articles and review papers, and 680 documents were selected at this stage, as shown in Figure 1. In the next step, articles from 10 ASEAN countries were selected. The last literature search ended up with 65 documents from Scopus and 53 from all other databases. That made a total of 118 papers to be considered for further review and the implementation of inclusion and exclusion criteria. The data was then exported to an Excel sheet to continue the systematic review. We also removed three documents from the Excel sheet due to full articles not being in the English language. Figure 1 shows the PRISMA framework implementation in this review.

### 2.1. Quality Evaluation

This review covers published originals and review papers to find the best outcomes and an excellent overview of the earlier data. Results, abstracts, and conclusions were separated to limit the archives. Additionally, cited references in the evaluated articles were also considered. The records were checked many times to avoid duplication, and to improve the desired results, irrelevant studies were also removed.

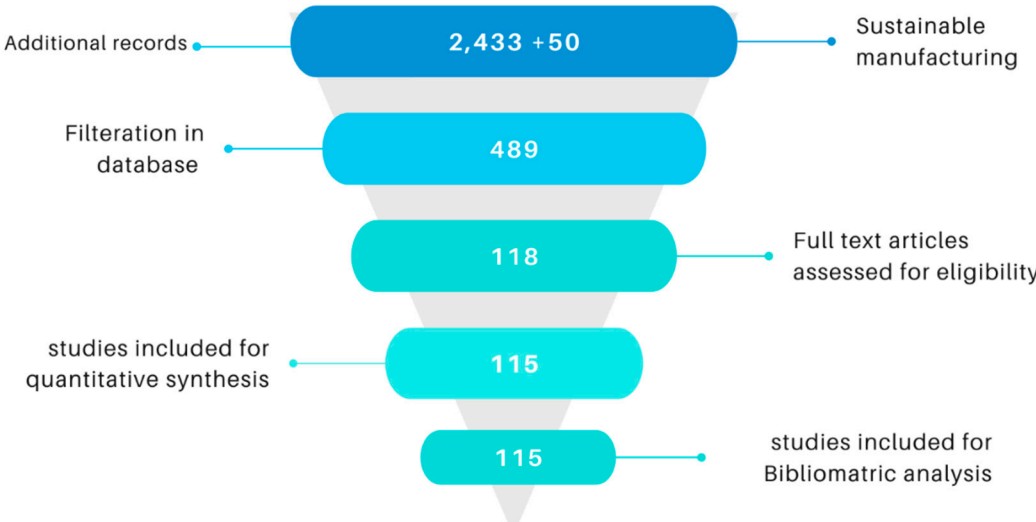

**Figure 1.** PRISMA framework.

## 2.2. Eligibility and Inclusion

The following selection criteria were adopted among the identified research articles: To ensure more accurate selection, research articles in the English language were selected. Furthermore, documents were selected provided they were published in the Web of Science, Scopus, ProQuest, EBSCO, or Science Direct databases. These databases are more reliable and better structured than Google Scholar or any other search engine.

This article reviews sustainable manufacturing literature and gives an overview of past studies in ASEAN countries and provides an agenda to improve manufacturing processes. Most research included quantitative results. Some studies presented a qualitative analysis of the high value that was incorporated and a few studies were based on review papers.

## 2.3. Studies Included in Qualitative Synthesis

After selecting the documents, a process composed of two consecutive steps was utilized to ensure the quality of analysis performed on the selected papers. Initially, the consistent metadata was imported into Microsoft Excel to perform a descriptive analysis of sustainable manufacturing literature, such as the qualitative and quantitative work done in ASEAN countries. In the following phase a thorough content analysis was performed to classify and examine key investigation streams, reporting recent research across diverse topics and emphasizing potential issues and chances for upcoming study. Content analysis is a research approach to analyze documents and texts that seek to describe and measure the manifest communication content about prearranged groups following a systematic method, permitting replicable and valid implications of texts.

## 3. Results

Figure 2 demonstrates the evolution of the number of publications per year on sustainable manufacturing from ASEAN countries during the last decade. The idea was to find sustainable manufacturing adaptations and procedures from the published literature. ASEAN countries were very keen to create better environmentally friendly businesses and industries to clean the atmosphere. As time passed, research in this area developed progressively between 2015 and 2020, reaching a peak in 2020. Figure 1 shows that most work on sustainable manufacturing was done in the last five years. Figure 1 shows that in the years 2016–2018, there were 15 papers each year. The year 2015 also has a score of 14 articles. However, the previous years' productivity in research literature was very limited. The year 2014 only had four studies on the topic of sustainable manufacturing.

Sustainable manufacturing is a relatively new concept for the manufacturing industries in ASEAN member countries.

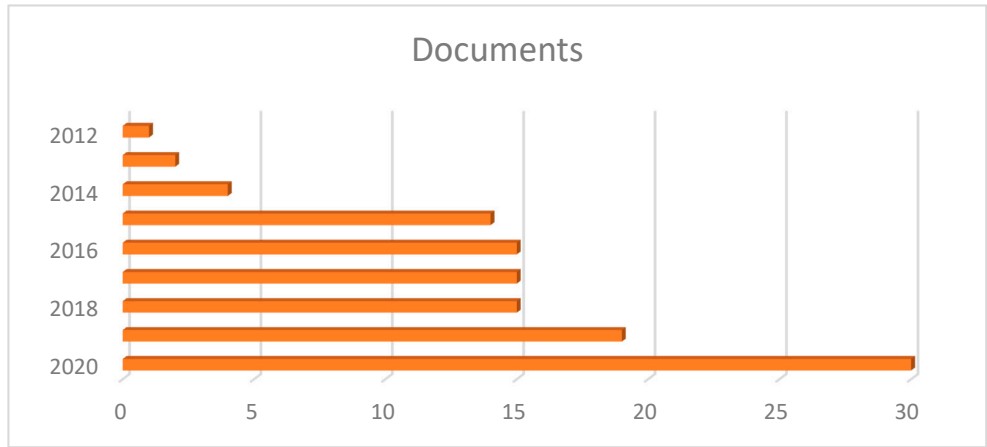

**Figure 2.** Distribution of published documents from 2011 to 2020.

The literature from the last 10 years included publications on sustainable manufacturing. Malaysia produced the highest number of studies on sustainable manufacturing, with 38 published records. Malaysia's work on a sustainable environment is notable compared to the other ASEAN countries. Singapore and the Philippines did the second most work, with 10 studies from each country on sustainable manufacturing. Very a few studies on the sustainable manufacturing field were focused in the remaining part of the ASEAN members. Brunei's, Cambodia's, Vietnam's, and Myanmar's progress in sustainable manufacturing research studies was meager during the last decade. Figure 3 shows the country-based literature publication list.

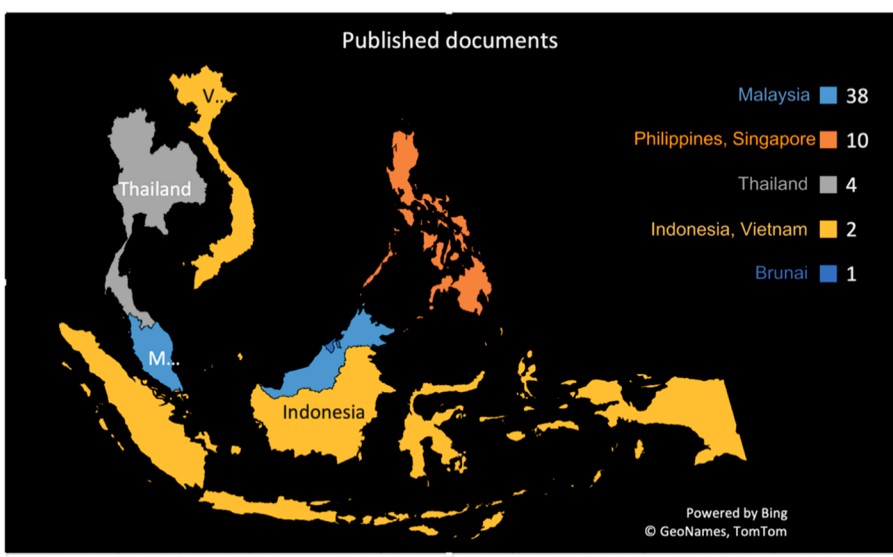

**Figure 3.** Distribution of published documents among ASEAN countries (2011–2020).

The subject categories were identified from the Web of Science database, and the results are indicated in Table 1. Most of the literature was about manufacturing engineering, green and sustainable science and technology, environmental sciences, and mechanical engineering, comprising 13%, 12%, 10%, and 9% of the total documents included in the review, respectively. The distribution of published research according to discipline is given in Table 1.

**Table 1.** Distribution of published records according to research discipline (2011–2020).

| Web of Science Categories | Records | Percentage |
|---|---|---|
| Engineering, Manufacturing | 15 | 13% |
| Green & Sustainable Science & Technology | 14 | 12% |
| Environmental Sciences | 12 | 10% |
| Engineering, Mechanical | 10 | 9% |
| Engineering, Environmental | 9 | 8% |
| Engineering, Multidisciplinary | 9 | 8% |
| Materials Science, Multidisciplinary | 8 | 7% |
| Management | 9 | 8% |
| Chemistry, Multidisciplinary | 3 | 3% |
| Engineering, Chemical | 3 | 3% |
| Engineering, Industrial | 3 | 3% |
| Automation Control Systems | 2 | 2% |
| Computer Science, Artificial Intelligence | 2 | 2% |
| Environmental Studies | 2 | 2% |
| Multidisciplinary Sciences | 2 | 2% |
| Operations Research & Management Science | 2 | 2% |
| Thermodynamics | 2 | 2% |
| Biotechnology & Applied Microbiology | 1 | 1% |
| Business | 1 | 1% |
| Computer Science, Interdisciplinary Applications | 1 | 1% |
| Energy & Fuels | 1 | 1% |
| Humanities, Multidisciplinary | 1 | 1% |
| Materials Science, Biomaterials | 1 | 1% |
| Materials Science, Composites | 1 | 1% |
| Metallurgy & Metallurgical Engineering | 1 | 1% |
| Total | 115 | 100% |

The citation report of the studies in the 10 years from 2011 to 2020 is presented in Table 1. The most-cited journal was the Journal of Cleaner Production, with 62 citations. The article name is "A weighted fuzzy approach for product sustainability assessment: a case study in the automotive industry" [23]. The article received most of its citations in the year 2018, with 12 citations that year. After that, the second most-cited study was "The need for global coordination in sustainable development" [24], cited 49 times in 10 years. The study was cited seven times in the year 2011 and published in the Journal of Cleaner Production. The article titled "Fuzzy-based sustainable manufacturing assessment model for SMEs" [25] was the third most-cited study in the last 10 years, with 31 citations. The study was published in Clean Technologies and Environmental Policies and cited 11 times in 2018. The fourth study was "A hybrid group leader algorithm for green material selection with energy consideration in product design [25]." It was cited 27 times in 10 years and published in CIRP Annals–Manufacturing Technology. There are some other cited studies shown in Table 2.

**Table 2.** Most-cited articles.

| Title | Authors | Source Title | Total Citations | Average per Year |
|---|---|---|---|---|
| A weighted fuzzy approach for product sustainability assessment: a case study in the automotive industry | Ghadimi et al. [23] | *Journal of Cleaner Production* | 62 | 7.75 |
| The need for global coordination in sustainable development | Jegatheesan et al. [24] | *Journal of Cleaner Production* | 49 | 4.45 |
| Fuzzy-based sustainable manufacturing assessment model for SMEs | Singh et al. [26] | *Clean Technologies and Environmental Policies* | 31 | 5.17 |
| A hybrid group leader algorithm for green material selection with energy consideration in product design | Ta et al. [25] | *CIRP Annals – Manufacturing Technology* | 27 | 6.75 |

**Table 2.** *Cont.*

| Title | Authors | Source Title | Total Citations | Average per Year |
|---|---|---|---|---|
| The impact of sustainable manufacturing practices on sustainability performance: Empirical evidence from Malaysia | Abdul-Rashid et al. [10] | *International Journal of Operations & Production Management* | 24 | 8 |
| Strategy selection for sustainable manufacturing with integrated AHP-VIKOR method under interval-valued fuzzy environment | Singh et al. [27] | *International Journal of Advanced Manufacturing Technology* | 21 | 5.25 |
| A finite element based data analytics approach for modelling turning process of Inconel 718 alloys | Vijayaraghavan et al. [28] | *Journal of Cleaner Production* | 13 | 3.25 |
| Integrating Axiomatic Design Principles into Sustainable Product Development | Beng et al. [29] | *International Journal of Precision Engineering and Manufacturing – Green Technology* | 13 | 2.17 |
| Design for environment and design for disassembly practices in Malaysia: a practitioner's perspectives | Ghazilla et al. [30] | *Journal of Cleaner Production* | 11 | 2.2 |

The journals that published the most articles on sustainable manufacturing in the ASEAN region are reported in Figure 4. The *Journal of Cleaner Production* and the *Journal of Technology* are at the top of the list with 17 and 10 publications, respectively. The *Journal of Materials* is the third largest on the list with five studies, along with *Sustainability* with the same number of published articles on sustainable manufacturing. The rest of the list is also shown in Figure 4.

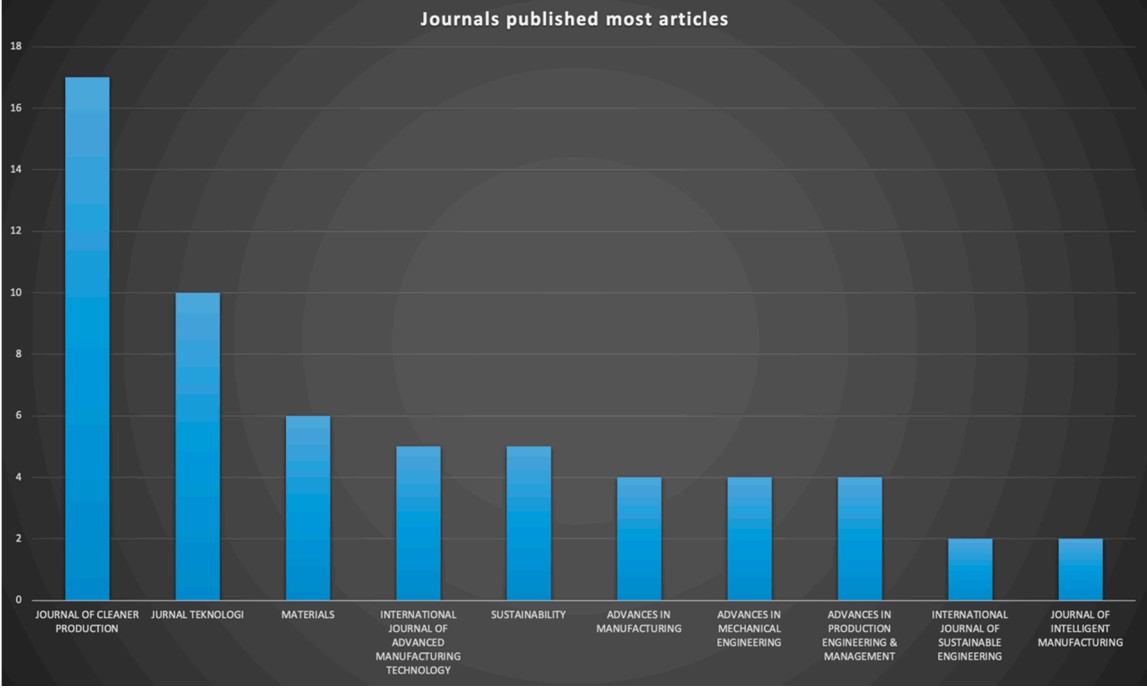

**Figure 4.** Journals with the most published articles.

## 4. Classifications of Literature on Sustainable Manufacturing Practices

The studies were further analyzed through content analysis to explore the classifications of the research. VOSViewer software was used to analyze the content of the published articles. Data networks

based on the text were created to cluster the linked concepts. Recent research confirmed that author keywords and keywords added in the process of the publications' indexation in the databases are equally effective for bibliometric analysis aimed at exploring the structures of research fields [31]. Therefore, we employed both categories of keywords for the co-occurrence analysis within the research field related to sustainable manufacturing in ASEAN countries. In total, 115 records were included in the research, and the data provided 2401 keywords. We carefully refined and selected only the most frequent 241 keywords repeated in a minimum of 10 documents. Figure 5 indicates the results of the content analysis. The network showed three major clusters, represented in different colors in Figure 5. The cluster represented in green shows the research on the sustainable product development process. The cluster in blue is mainly attributed to environmental management practices in the production process. Finally, the red cluster indicates sustainable manufacturing performance-related research. Each cluster is further analyzed in the next subsections to identify the manufacturing industry's current practices.

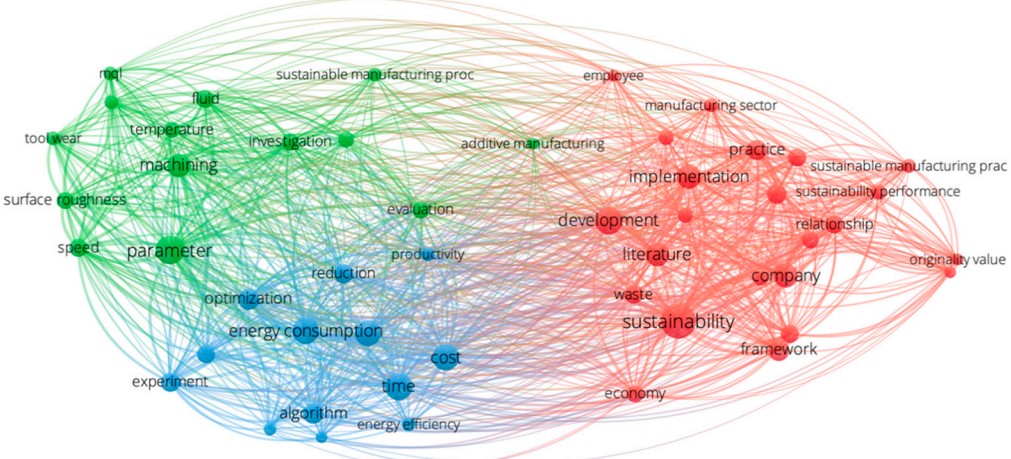

**Figure 5.** Research classifications using the text network technique.

### 4.1. Sustainable Product Development Process

The major significant contribution to our review was from the product category from the classification of data we found. The product development process is an essential element in the manufacturing process. The research made a significant contribution to the different stages of the overall product life cycle. We found that most data is relevant to the product development stage of the manufacturing process. Nugroho and Zhu [32] focused on sustainable supply-chain planning development for the biofield platform. The study aimed to design a network that developed the product and production process's supply-chain process. Findings from the research suggest that biofuel platform planning and product distribution significantly influence economic growth, reducing environmental impacts and helping local community development prosper. Another study discussed public and private partnerships in Singapore. The pharmaceutical industry collaborated with the university to counter the industry's challenges in getting sustainable manufacturing. The collaboration's focus point was to develop innovative, interdisciplinary solutions to achieve sustainable manufacturing in the pharmaceutical industry in Singapore [33,34]. These cross-industry collaborations helped sustainable product development in ASEAN countries.

The literature concerned the development of the sustainable manufacturing of products in ASEAN member countries. Ingram et al. [35] highlighted some manufacturing approaches during the development phase, along with equipment—such as using a different kind and amount of magnetic material. The material flow continues during the product life cycle. However, the environmental impact is also ongoing throughout the product life cycle: processing, manufacturing, utility, disposal, and supply chain. Product life stages are usually crucial during the production process. Triple bottom

line (TBL) theory is at the core of the manufacturing process to evaluate the overall process [36]. Triple bottom line (TBL) consists of economic, environmental, and social pillars that are developed to measure a firm's performance.

The sustainability evaluation process represents indicators that show the economic, environmental, and social parts of the TBL in the literature [37,38]. According to Ahmad et al. [39], environment, economy, and society influence sustainability in the manufacturing sector. Findings from the study were critical for the environmental category, with less use of solid waste and more frequent use and development of material, energy, and air emissions. For a sustainable manufacturing process, the process design is crucial to the product life cycle [40]. Product design classification leads to the next category about experimental research in ASEAN member countries. Two sustainability-related pieces of empirical literature revealed that the Fuzzy Analytic Network Process (FANP) is being used widely to identify the infrastructural decisions that support sustainable manufacturing that can be integrated with traditional manufacturing practices. FANP was also used to determine the optimum firm size for the integration of sustainable approaches to classical manufacturing. The study found that firms need to adopt requirements for products that must be ordered to make the production system.

Large firms need to develop the decentralized operational areas, but the system is integrated for better communication of production-related processes. It also stresses better social skills to support active and functional regions [41]. Ghani et al. [42] contributed to the literature; their research explored the developments of heat treatment applications to achieve sustainability in manufacturing. Some industries have developed a process that is less detrimental to the environment. For example, the metal-cutting industry uses techniques like green machining, which is a metal-cutting process that does not infiltrate any fluid, limiting liquid waste.

Similarly, Iesa et al. [43] emphasized forming methods and heat treatment to reduce the environmental impact of manufacturing. For sustainable manufacturing, the experimental research in different segments of the industry and internet-based contributions were discussed. These recent developments are crucial in the manufacturing industry to mitigate waste and greenhouse gas emissions. Table 3 indicates the literature on sustainable product development.

In the life cycle of a product, energy consumption evaluation and analysis are critical issues for achieving green and sustainable manufacturing. A very recent approach urged the emphasis of the employment of the Internet of Things (IoT) and cloud-based techniques to measure energy consumption [61]. These new technologies are essential for measuring and, to some extent, controlling the environmental impact of manufacturing and have also drastically improved manufacturing processes [62]. However, experimental studies discuss mass production areas and industries for sustainable manufacturing, and according to Vijayaraghavan and Castagne [63], the material utilization and energy consumption during the manufacturing process are diverse parameters mainly related to input and processes—for example, throughput time, media used, material used for the workpiece, and cycle times. All of these process parameters are measured through a trial-and-error basis mostly in the manufacturing industry. This results in inefficient utilization of resources, enhanced waste, and energy inefficiency. However, recycling can be considered a potential alternative to reducing waste and reusing it. However, the recycling process itself is resource intense [64]. Remanufacturing in the era of Industry 4.0 can increase efficiency, reliability, and digitalization of the remanufacturing process [65]. Remanufacturing is a crucial enabler for sustainable products due to its effectiveness in closing the loop on material flows, extending the product life cycle, and reducing production waste and emissions [65]. With that, some literature enlightened the process of the recovery of natural resources within the sustainable manufacturing process [66].

**Table 3.** The literature on sustainable product development.

| Authors | Classification | Settings | Procedures |
|---|---|---|---|
| Majeed et al. [44] | Smart manufacturing | Sustainable manufacturing | Product life cycle |
| Jain et al. [45] | Environmental regulations | Auto manufacturers | Design phase |
| Ordoñez Duran et al. [46] | Product design | Environmental impact | Sustainability indicators |
| He et al. [47] | Product design | Natural dynamic control systems | Robotic ankle |
| Zheng et al. [48] | Defective products | Design phase | Emission reduction |
| Ingarao et al. [35] | Product life cycle | Production | Environmental impact |
| Gunna R V. [49] | Product life cycle | Triple bottom line (TBL) | Production schedules |
| Kamalakkannan et al. [50] | Product life cycle | Environmental sustainability | Tea industry |
| Jegede et al. [51] | Energy consumption | Eco-impact | Stainless steel |
| Bevilacqua et al. [52] | Assessment | Environmental impact | Welding speeds |
| Üstündağ Okur et al. [53] | Design | Product demand | Solar energy |
| Laverne et al. [54] | Design | Environmental impact | Electrical energy |
| Ansari and Modarress [55] | Design | Environmental impact | Motor vehicles |
| Zhu et al. [56] | Development | Design of biofuel supply chain networks | Biofuel |
| Raoufi et al. [57] | Development | Design | Software tools |
| Balakrishnan et al. [58] | Eco-friendly product | Environmentally responsible | Global warming |
| Miranda et al. [59] | Development | New-generation products | Marketplace |
| Ahmad et al. [39] | Design | Environmental impacts | Aluminum |
| Ahmad et al. [38] | Assessment | Triple bottom line (TBL) | Environment, economy, and society |
| Yu et al. [60] | Green products | Environmental policies | Supply chain |
| Yu et al. [60] | Design | Multi-criteria decision making | Eco-efficiency |

*4.2. Environmental Management and Monitoring Regulations*

Environmental management and regulation consist of planning, assessment, implementation and direction, and implementation tools that ensure the enforcement of laws and policies. Although integrated environmental management is the most widely promoted global environmental management practice used in sustainable management, it has yet to manifest extensively in firms' ethics, especially when dealing with water, land resource, and ecology issues. Regulations to control the internal environment and external environment in ASEAN countries are primarily established in Malaysia and Singapore, showing the efforts made by countries with extensive industry and management issues. Jain and Hazra [67] conducted a study on auto manufacturers' design decisions that impact environmental regulations. Government regulation-implementing authorities determine greenhouse gas emissions from vehicles manufactured by different auto producers. The study's findings showed that when customers have low price sensitivity, it creates competition for manufacturers to increase vehicle production and regulators apply strict actions. The sustainable manufacturing system's prior development process is recognized as a highly useful tool to decrease environmental impact during manufacturing. Nujoom et al. [68] added that the lean approach is also reliable for achieving sustainability that improves productivity with efficiency in the system and minimizing waste. However, the lean approach's problem is not including environmental waste such as energy consumption and $CO_2$ emissions. The findings of the study pointed out that each process uses several machines and other energy consumption units. According to the International Organization for Standardization (ISO) regulations, many industries use and consume energy at a maximum level. However, material processing industries need to decrease energy and carbon emissions [69]. According to Muñoz-Villamizar et al. [70], their study findings show that energy and material processes are firmly centered on environmental sustainability. Environmental protection due to the implication of sustainable manufacturing practices is vital. Table 4 shows literature on environmental management and monitoring regulation studies.

**Table 4.** The literature on environmental management and monitoring regulations.

| Authors | Classification | Settings | Procedures |
|---------|----------------|----------|------------|
| Lu et al. [71] | Corporate social responsibility (CSR) | Small- and medium-sized enterprises (SMEs) | Policies |
| Geda et al. [72] | Environmental regulations | Vehicle | Greenhouse gas emissions |
| Ordoñez Duran et al. [46] | Regulations | Environmental Performance Index (EPI) | Energy consumption |
| Han et al. [73] | Producer's responsibility | Environmentally friendly recovery | Recovery decision support system |
| Meng et al. [74] | Environmental regulations | Metal-cutting industries | Eco-friendly techniques |
| Sivaiah and Chakradhar [75] | Government interventions | Social welfare | Green products and non-green products |
| Gao et al. [76] | Rules and regulations | Manufacturing industries | Energy saving and minimizing the production of carbon dioxide |
| Kamalakkannan et al. [50] | Government environmental regulations | Small- and medium-sized enterprises (SMEs) | Environmental effects |
| Gao et al. [76] | Production principles | Sustainable manufacturing practices | Energy and material conservation |
| Choi and Lee [77] | Environmental rules | Globalized market | Green performance measures |

Environmental management efforts include interrelating green purchasing management, reverse logistics, product stewardship in the supply chain, and product and process innovations [68–80].

### 4.3. Sustainable Performance

Large-scale industries and manufacturing create problems for the internal environment of the organizations and the external natural environment. The manufacturing process is harmful to natural resources and the environment [80]. Development at the cost of the natural environment is stressed in literature around the world. Moving towards sustainability in current literature is widely embraced by researchers. The key is to maintain the equilibrium between economic activities and environmental damage [81]. Sustainable manufacturing practices are significantly contributing to environmental protection. At the same time, these technologies are providing economic benefits to industry and socially acceptable. The development of environmentally friendly practices and the removal of adverse techniques is the goal of intelligent systems and process improvement for sustainable manufacturing. However, "environmental performance is challenging from the decision-making perspective because of difficulty determining and prioritizing the proper factors that have a significant effect on a firm's environmental performance" [82]. According to Rezai et al. [83], awareness about environmentally friendly products has drastically increased, and consumers are increasingly demanding products that are not only environmentally friendly but also have a circular life cycle.

The manufacturing industry's recent development suggested adopting advanced manufacturing technology (AMT) and cloud manufacturing practices to enhance manufacturing capabilities, develop sustainable manufacturing processes, and mitigate environmental effects [84]. In the current world, growing environmental issues lead new product development to be altered to adopt environmentally friendly practices for sustainable manufacturing [84]. When such advanced techniques adopt environmental sustainability, the benefits increase manufacturing capacity, improve the industry's image in regard to environmentally friendly products, and enhance business profile, consumer perceptions, and corporate reputation [85]. Nevertheless, many manufacturing industries are still unable to correctly implement sustainable manufacturing practices [86,87]. For environmental sustainability, there is a need to improve social and ecological sustainability. One study discussed small- and medium-sized enterprises in Malaysia. According to Hami et al. [88], small- and medium-sized enterprises (SMEs) play a significant role in Malaysia's economic development, accounting for approximately 97.3% of businesses in the country. Therefore research on environmental and social sustainability needs to be increased. In environmentally sustainable manufacturing processes, friendly production also highlights

energy consumption during the production process. Most of the recent efforts in the manufacturing industry are focused on improving the energy efficiency of the manufacturing process. However, little attention has been paid to a holistic approach to manufacturing efficiency [89].

To control the environmental impact of manufacturing processes, industries are wisely working on energy-efficient manufacturing with end-of-life resource utilization management and waste minimization. The concept of sustainable manufacturing is being practiced widely in developing nations. However, the implementations of sustainable manufacturing practices are in their infancy in developing countries [84]. The literature from the Malaysian perspective in one study assessed the decision-making processes on possible changes in policies and operational procedures for implementing sustainable manufacturing practices in their organization. In the environmental section, most of the literature leads to the Malaysian manufacturing industries, and environmental concern is shown in large manufacturing industries such as automotive and metal base producers. Still, the contribution from other ASEAN member countries is almost significantly less. Researchers from the other member countries also need to realize the importance of the manufacturing industries' environmental challenges regarding environmental sustainability [90]. In the next section, the collaborative research toward sustainability between ASEAN countries is discussed.

## 5. Bibliometric Analysis of Collaborative Research and Development in the ASEAN Region

The second objective of this paper was to analyze the collaborative research on sustainable manufacturing in ASEAN countries. A bibliometric analysis was done to determine the collaborative research in these countries. The analysis segmented the research collaborations into three main aspects. First, the clusters represented in Figure 6 indicate the most dominant regions with strong internal and external partnerships. The figure depicts Malaysia leading the area in terms of external research collaborations, followed by Singapore and Indonesia.

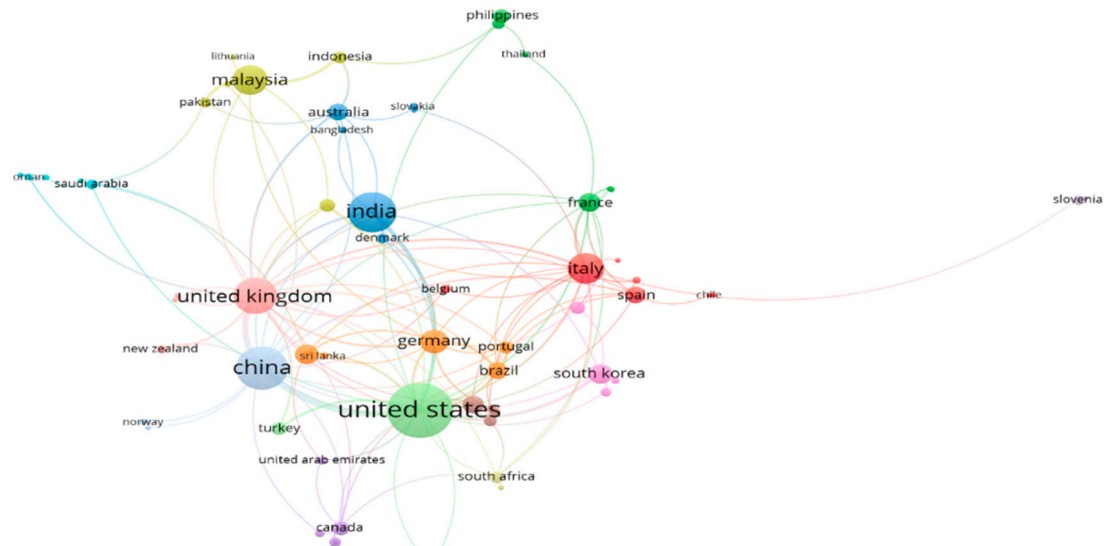

**Figure 6.** Collaborative research in ASEAN countries.

This indicates that these three countries are putting strong efforts into mitigating the production process's adverse effects on the natural environment and resources. A multidisciplinary review was done collaboratively between Malaysia and the USA to review zero-waste manufacturing in those countries and sustainable manufacturing [91]. However, the collaboration patterns changed over the year. Figure 6 indicates the research collaboration patterns from the past five years. The colors in Figure 6 show the trend of research collaborations in the region. The more similar the countries' color is, the more likely it is that those countries are collaborating. It was observed that until 2014, the collaboration activities in the ASEAN region, mainly in Malaysia, Singapore, and Indonesia,

were more dominated towards developed countries such as the United States, the United Kingdom, and Australia. These countries are clustered in dark colors in Figure 6. Figure 6 indicates that ASEAN countries are more open to developing countries for collaborations. This shift may bring technological reforms. The increase in collaborative research between ASEAN and Asian countries can bring low-cost solutions to the region and ease resource sharing. On the other hand, collaborations with Sweden, the Czech Republic, Hong Kong, South Korea, the UK, and the USA can lead to the latest technological breakthroughs to counter sustainability challenges.

Finally, it is equally important to assess the efforts made by each country in the ASEAN region in forming collaborative research networks. Figure 6 highlights the number of joint activities carried out by each country and indicates that Malaysia dominates the region with the highest number of internal national collaborations with developed and developing countries. In contrast, Singapore is more focused on developing countries. Indonesia is trying to maintain cooperation with both Singapore and Malaysia.

## 6. Conclusions

Summing up the conducted literature analysis allows us to answer the research questions of the study. The systematic literature review and bibliometric analysis were focused on two significant objectives. The first was to map existing sustainable manufacturing practices in ASEAN countries. To achieve the first objective, we reviewed environmental sustainability work. Past research showed quality work in environmentally friendly practices in ASEAN countries, especially manufacturing procedures and supply-chain channels. Waste minimization, remanufacturing, and recycling procedures were also part of the review. Lean management for sustainable manufacturing processes and improvement in delivering products on time was also considered. Furthermore, these concepts were classified into the related cluster. Most of the past research was focused on new product development, and stage-gate methods were employed to assess manufacturing efficiency and resource utilization. Due to data and other resource requirements, only a few studies were found with extended boundaries. However, studies were reported from the ASEAN countries; most research was done in Malaysia, and the focus of the studies was sustainable manufacturing in Malaysia. Figure 7 shows the mapping of sustainable manufacturing practices in the past decade.

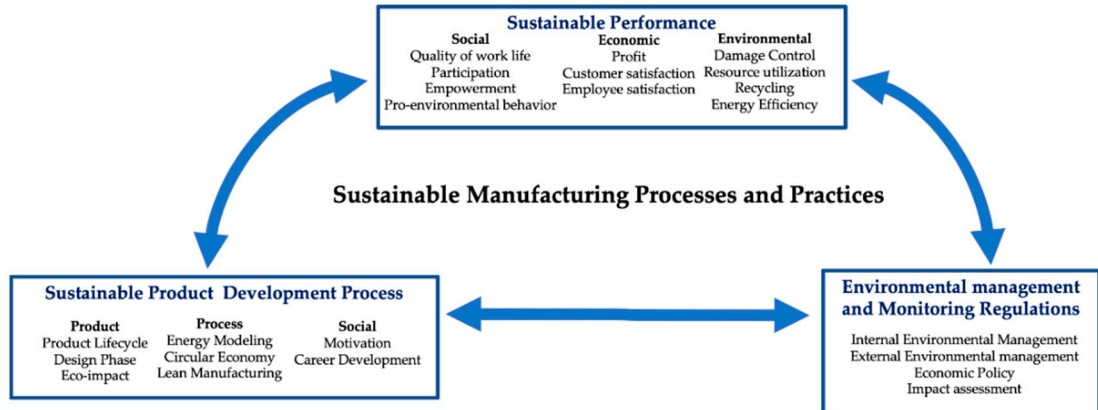

**Figure 7.** Sustainable manufacturing processes and practices.

The research focused on sustainable product development processes was mainly attributed to the product, process, and social sides of the product development process [92]. Environmental management and monitoring regulations were concerned with environmental management, economic policy, and impact assessment [93]. On the other hand, studies related to sustainable performance focused more on sustainability's social, economic, and environmental aspects.

The second objective of this research article was to assess the collaborative research work among ASEAN countries to achieve the common goal of green growth. Bibliometric analysis using country co-occurrence of the authors was used to analyze each country's role in collaborative efforts. The results indicate that Malaysia is prominent in establishing research collaborations among ASEAN countries. Most other countries focus on research collaborations with developed countries. A limited number of studies was found from Singapore, which had coauthors from ASEAN countries. Therefore, research collaborations are inadequate among ASEAN countries. In order to develop low-cost sustainable manufacturing solutions, it is crucial for ASEAN countries to enhance collaborative efforts and resource sharing.

*Literature Gaps and Future Agenda*

The findings indicate that sustainable manufacturing research is concentrated on products, plants, and processes. However, manufacturing at the sector level and its economic effect are not studied often. More effort is required to get industries from other parts and ASEAN countries on board to develop their sustainable manufacturing performance comprehensively. The study's findings highlight that sustainable manufacturing is a holistic approach and needs to be improved in all aspects of the manufacturing process. For this reason, manufacturers should think and analyze beyond product development and stage-gate processes to achieve three objectives of sustainable manufacturing. These objectives are labeled as the triple bottom line (TBL) (i.e., environmental performance, economic performance, and social performance). The TBL concept is discussed in a minimal number of studies; researchers must work on TBL and technology's advanced concepts to improve the quality of manufacturing.

Most of the work in this regard has been carried out in Malaysia, while the other ASEAN member countries' contribution is meager. Singapore and the Philippines have also contributed some work, but the other members' productivity in the literature is very low.

This study found some new concepts such as the Fourth Industrial Revolution (IR 4.0) in some literature and advanced manufacturing models for sustainability [94]. Researchers must focus on hybrid processes and IR 4.0 to establish sustainable manufacturing procedures in ASEAN member countries. More efforts are also needed to make social and economic assessments more effective and applicable to manufacturing industries.

Environmental issues in the ASEAN region are getting severe, and climate change poses a severe threat to these nations. Climate change is causing disasters that humankind has never met before. That is a serious concern over the years, and the manufacturing industry is most responsible for environmental degradation. For this reason, a massive amount of research has taken place in the past decade to develop manufacturing practices that can reduce environmental damage and resource consumption. However, countries in the ASEAN region need to work closely with each other due to similarities in issues faced by the region. Although collaborative research efforts have increased, these efforts are mostly directed towards collaborations with other regions. Collaborations within the ASEAN region are scarce.

**Author Contributions:** This work was the output of collaborative research between researchers from different nationalities and institutions. N.K. and M.I.Q. were responsible for the conceptualization of the idea, manuscript preparation and data analysis, and revision of the manuscript. The prepared manuscript was reviewed and amended by T.R., S.Q. and S.M. contributed to the revised manuscript and also secured article processing charges to facilitate publication of the research article. S.S.H. helped with reviewing and proofreading the final manuscript. All authors have read and agreed to the published version of the manuscript.

**Funding:** This research received no external funding.

**Acknowledgments:** This work was a result of collaborative research between researchers from the UniKL Business School, Universiti Kuala Lumpur, Malaysia; the Faculty of Technology Management and Technopreneurship, Universiti Teknikal Malaysia Melaka, Malaysia; the Institute of Applied Psychology, University of the Punjab, Lahore, Pakistan; and the Department of Gender and Development Studies, Lahore College for Women University, Pakistan.

**Conflicts of Interest:** The authors declare no conflict of interest.

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
