# Peer review of "Classifications of Sustainable Manufacturing Practices in ASEAN Region: A Systematic Review and Bibliometric Analysis of the Past Decade of Research"

_sustainability, doi:10.3390/su12218950_

Round 1
Reviewer 1 Report
I would not accept this paper as it is now, since the methodology is not very accurately explained and the results are not clearly presented (tables, graphics, ) to be lately explained in their place. Apart from that, I cannot see the provided conclusions giving a clear response to the set objectives, and the drafting of the paper should be much improved. There is not any classification of the manufacturing practices concluded...
Some other comments and suggestions are:
Third affiliation is missing
Line 19. INFect??? There should be a mistake. Additionally, starting with minor letters…
Line 41-44. References are needed.
Line 47. Martial?
Line 48-49. The concept of linear economy should be somehow referenced in the doc.
Line 51-54. References are needed.
Line 63. There should be a mistake. Please revise. “t”.
Line 74, 77 etc. Please correct CO2
Line 98. The reference should be located at the end of the sentence.
Line 100. This concept has been already repeated.
Line 106.the word magnitudes (26) does not make any sense there.
Line 109. The reference is badly written.
Line 117-118. This sentence is weird. It should be drafted again.
Line 122. Which is the ASEAN declaration?
Line 123. These are the two main objectives of this paper.
Line 131. Are authors refereeing to scientific journals?
Line 133. The verb is missing?
Line 139-145. In my view, this paragraph is a bit confusing. English and drafting must be revised.
Line 146. 115 twice should have the same size in the figure
lIne 147. Fig 1 RISMA or PRISMA??
Line 151. The papers were not evaluated within this research but studied, weren’t they?
Line 153. What were the reasons to identify paper as not relevant? Please, specify.
Line 163. In my view, the whole paragraph needs to be drafted again.
Line 164. The paper refers to 65 papers, but I wonder how did authors come up with this amount of papers?
Line 179. A batter?
Line 182-185. This sentence does not make any sense.
The last paragraph, located before Fig 2 needs to improve the English drafting.
Fig 2 . the Y exe needs to identify what is this graphic measuring, and in the X exe, the names of the Countries should be better written..capital letters…more separate..etc.
Fig 2. And Fig 3. And table 1 (2011 to 2021). 2021?? We are living in 2020. Table 1 refers to 2012 instead of 2011 (set in the figures).
Fig 3. The Y exe refers to the number of papers??
Fig 3. How many ENGINEERING fields are? MATERILS SCIENCE?
Line 230 The acronym SME and ASEAN should be explained first time (Small and medium-sized enterprises …)
Fig 4. Many typos in this graphic. JOURNAL TECHNOLOGY instead of what is written in the paper.
Additionally, after Fig 4. Capital letter should be placed.
Line 243. It says that the studies are classified after analysing the excel sheet according to the process. But the process, in my view is not very accuracy and sound based.
Line 242. I am quite confused with this 4.0 point since it seems it not providing results but stating and identifying categories.
Line 251. From which table or data authors can say the statement in line 250-251?
Line 255. The study aims again??? We are in the results part.
In a general level, in my view, data obtained by this research should be clarified and much improved.
Table 2. Sometimes the references are in brackets and others in parenthesis. Please, be coherent.
I cannot see the difference in between the classification and the procedures.
Line 308. Throughput??
Line 339. The statement concerning ISO needs a reference.
Line 345. Table 3. Please order the number of the references
Line 367. batter environmental effects?
Lines 370-372. Authors stated this…but it should be somehow taken from the review. Which authors are saying that?
Line 386-387. Are these statement results of this review? From which data? Data are missing
Line 399. In this line the most important objective is depicted, but it is not the same stated before for instance in the abstract (mapping the existing sustainable manufacturing literature). Please clarify.
Line 406. Fig 5. Some names are very small. Illegible.
Line 412. I cannot understand what the colour of fig 5 mean.
Line 423. Fig 6. Trends? Which trends? I can imagine the number because of the size of the coloured extension but not the trends.
Line 467-468. The new researchers have to have the opportunity to explore the concern fields for improvement in the literature…is this a real conclusion of the review?
References.
Line 505. The year is missing.
Author Response
Dear Reviewer,
we have corrected the manuscript based on your valuable comments. each point was considered and amendments were done in the manuscript accordingly. the detailed correction report is attached. and also highlighted in yellow in the manuscript.

Reviewer 2 Report
This review is interesting but could be improved with the following considerations:
It would be better if you improve your writing. I recommend having a native English speaker to review your article. Such as lines 427 and 457.
In section 4.0 it would be convenient to include what articles and what scope they had around new technologies or Industry 4.0, since it talks about IoT, I4.0, cloud and is not specified.
The review reflects the publications that exist, the topics covered and the situation regarding the field of sustainable manufacturing in ASEAN countries, but the gaps that exist are not clearly stated. Furthermore, it has to clearly set a new and promising research agenda for the future in the ASEAN region.
Finally, there are some comments about the figures:
Comments on figure 2:
The reference that is made in line 182 about 33 summits in Singapore was not with the same number reflected in the graph. In addition, when using line 175 to comment on the evolution of existing publications per year, it would be better to use a graph in which time appears as one of the axes to clarify understanding for the reader. On the other hand, the horizontal axis on which the countries are located is difficult to read.
Comments on figure 4: why are journal cleaner production and jurnal teknologi in the figure twice? This figure does not enhance the information given by the text.
Author Response
Please find attached the correction list. the manuscript has been amended as per your recommendations.

Reviewer 3 Report
The article is correct, but it is written chaotically in several places (not to mention the editorial page, such as the title of figure 4).
From the main comments:
- no context of the analyzed data,
- the reasons for using this database during the research are too weak,
- no "discussion" part and references to the works of other authors in this research area,
- incorrect entry reference no 1, 33,
- less than 50% of the literature was written in 2016-2020.
Author Response
Dear Reviewer, please find the correction list based on your comments. all changes are highlighted in yellow in the manuscript.

Round 2
Reviewer 1 Report
Still, the structure of the manuscript does not correspond to the structure a scientific paper should follow.
Although, there has been much improvements in this second version, still, the conclusion of the paper does not fulfil the objectives of the paper. In my view, this is the main reason for my rejection of the manuscript like it is now. The conclusion seems to be kind of description. First objective of the paper has been achieved, but the second one, has not been reached. There is only a sentence at the end of the paper regarding this objective stated in the introduction. But there is not an explained methodology concerning this second goal, and results etc.
Minor suggestions:
Fig 4 should be fig 3…and then…the same with the followings
Fig 5 (fig 4) is illegible…this does not make any sense.
Lines 248 and 247 have different texting (paragraph structure).
Now, in this new version there are two table n. 2?
Please, avoid point number 4.0….There should be an introduction…of the point and then the subcategories…
Author Response
Dear Reviewer,
We have done corrections as per your recommendations. Kindly find the attached report.

Reviewer 3 Report
thank you for making changes to my and other reviewers. This makes the article better.
Author Response
Thank you so much for considering our paper. We have also done the minor changes suggested by the reviewer. here are the final corrections for your reference.

Round 3
Reviewer 1 Report
Still, conclusions should answer the established objectives. And i have not read any methodology answering the second goal, neither any conclusion.
Maybe authors should have to rewrite the objectives. Since they are not responding to the second goal: Assess the collaborative research work among ASEAN countries…
Author Response
We have provided methodology, results, and conclusion highlighted which deals with objective two of the study. I hope this serves the purpose. However, if it is still not satisfactory, we shall remove this part from the paper and work only the first objective.

Round 4
Reviewer 1 Report
This version of the manuscript offers a much more clear research. Thank you very much for having improved it.